SOFTWARE

# `ArborSim`: Articulated, branching, OpenSim routing for constructing models of multi-jointed appendages with complex muscle-tendon architecture

**Xun Fu[1], Jack Withers[2], Juri A. Miyamae[1], Talia Y. Moore** [ORCID][1,3,4]*

**1** Robotics, University of Michigan, Ann Arbor, Michigan, United States of America, **2** Computer Science, University of Michigan, Ann Arbor, Michigan, United States of America, **3** Mechanical Engineering, University of Michigan, Ann Arbor, Michigan, United States of America, **4** Ecology and Evolutionary Biology, Museum of Zoology, University of Michigan, Ann Arbor, Michigan, United States of America

* taliaym@umich.edu

**Data Availability Statement:** The source code of ArborSim is freely and anonymously accessible on GitHub, under the permissive Apache License 2.0,

## Abstract

Computational models of musculoskeletal systems are essential tools for understanding how muscles, tendons, bones, and actuation signals generate motion. In particular, the OpenSim family of models has facilitated a wide range of studies on diverse human motions, clinical studies of gait, and even non-human locomotion. However, biological structures with many joints, such as fingers, necks, tails, and spines, have been a long-standing challenge to the OpenSim modeling community, especially because these structures comprise numerous bones and are frequently actuated by extrinsic muscles that span multiple joints—often more than three—and act through a complex network of branching tendons. Existing model building software, typically optimized for limb structures, makes it difficult to build OpenSim models that accurately reflect these intricacies. Here, we introduce `ArborSim`, customized software that efficiently creates musculoskeletal models of highly jointed structures and can build branched muscle-tendon architectures. We used `ArborSim` to construct toy models of articulated structures to determine which morphological features make a structure most sensitive to branching. By comparing the joint kinematics of models constructed with branched and parallel muscle-tendon units, we found that among various parameters—the number of tendon branches, the number of joints between branches, and the ratio of muscle fiber length to muscle tendon unit length—the number of tendon branches and the number of joints between branches are most sensitive to branching modeling method. Notably, the differences between these models showed no predictable pattern with increased complexity. As the proportion of muscle increased, the kinematic differences between branched and parallel models units also increased. Our findings suggest that stress and strain interactions between distal tendon branches and proximal tendon and muscle greatly affect the overall kinematics of a musculoskeletal system. By incorporating complex muscle-tendon branching into OpenSim models using `ArborSim`, we can gain deeper insight into the interactions between the axial and

at https://github.com/EMBiRLab/ArborSim. The GitHub repository is called "ArborSim."

**Funding:** The author(s) received no specific funding for this work.

**Competing interests:** The authors have declared that no competing interests exist.

appendicular skeleton, model the evolution and function of diverse animal tails, and understand the mechanics of more complex motions and tasks.

## Author summary

OpenSim models of musculoskeletal systems are valuable tools for understanding the mechanisms underlying biological movement. However, highly jointed structures, like necks and tails, have been difficult to model, likely due to the presence of numerous bones, and their complex network of branched muscle-tendon units that span many joints. We introduce `ArborSim`, a modeling tool that facilitates construction of branched muscle-tendon units. Our comparisons of equivalent branched and unbranched models suggest that the modeling approach greatly affects the model predictions, which thereby indicates the importance of accurately modeling branching in multi-jointed structures. With `ArborSim`, we provide a unified framework that not only streamlines the creation of the complex musculoskeletal models of these structures but also paves the way for more in-depth investigations into the biomechanical behavior of branched muscle-tendon networks. This encourages more detailed study of the material properties and mechanics of complex muscle-tendon branching networks and highly jointed structures.

## Introduction

Animal movement results from intricate interactions among elements of musculoskeletal systems [1–3]. Investigating these interactions using experimental *in vivo* data alone can be challenging and often impractical due to measurement constraints and invasive methods. Computational musculoskeletal modeling and simulations offer a complementary approach. By integrating models of system elements, dynamic simulations utilizing the integrated model are generated, providing estimates of hard-to-measure variables [4, 5]. These simulations enable researchers to reproduce observed experimental results, comprehend system coordination, design innovative experiments, and predict treatment outcomes, making them valuable tools in biomechanics studies [4, 5].

In the last few decades, substantial advancements have been made in detailed musculoskeletal modeling of the human body [6–12], which has led to the discovery of fundamental human movement principles, advanced injury diagnosis, and informed prosthetic and robotic design [13–23]. OpenSim [24, 25], a widely-used open-source software for musculoskeletal modeling and simulation, has been a key contributor to these advancements, fostering the sharing and dissemination of musculoskeletal models among the biomechanics community through the SimTK database. Recently, there has been increasing interest in developing musculoskeletal models of other animals [26–31], such as ostrich [27], chimpanzee [26], and mouse [28, 31], with the aim of enriching the understanding of a wide range of forms and behaviors exhibited by other species, and inspiring the development of bioinspired robotic technologies [32–34].

In every vertebrate animal, highly jointed systems, such as phalanges and the axial skeleton (spine, including necks and tails), facilitate greater curvature and are crucial to performance and survival. For example, cats adjust the stiffness of their flexible backs to absorb kinetic energy upon landing, allowing their limbs to attenuate landing impulses more safely and effectively [35]. Certain monkeys use their prehensile tails as a fifth limb, providing essential grasp and balance while navigating through trees, facilitating climbing in their arboreal environment

[36]. As an extreme example, snakes are remarkable, highly jointed systems by themselves. They possess highly flexible and muscular bodies that allow for a range of movement adaptations, enabling them to effectively navigate various terrains and tight spaces [37–39].

Nevertheless, in most existing musculoskeletal models, highly jointed systems are often simplified or excluded to facilitate computationally challenging analyses [30, 40, 41]. For instance, a musculoskeletal model of the dog does not consider the motion of the spine as a series of individual vertebrae, but rather simplifies the spine into discrete regions such as thorax and tail which are granted three degrees of freedom (DOFs) each [30]. A mouse model simulating a trotting gait focuses only on the hindlimb, disregarding the spine and tail entirely [28].

The practice of simplifying highly jointed systems, while helpful in reducing computational complexity, may unintentionally mask their complete biomechanical importance. However, even within these limitations, recent studies using simplified, but articulated models of highly jointed systems have identified a significant role for them in animal locomotion. For example, using a musculoskeletal model of a dinosaur that includes a simplified two-link tail, researchers have found that the tail performs a critical role in enhancing locomotor efficiency [42]. Moreover, by incorporating tail compliance properties through a five-link tail model, researchers have used the natural frequency of a tail to estimate the preferred walking speed of a dinosaur [43]. These studies exemplify the unexpected and substantial contributions of highly jointed systems that await discovery through modeling, underscoring the potential for gaining even more profound insights from more detailed and comprehensive models of these systems.

Despite the development of musculoskeletal modeling software over the past few decades, constructing detailed models of highly jointed systems remains both time-consuming and challenging. The primary challenge results from the presence of numerous bones and intricate muscle and tendon architectures, in which muscles and tendons can span multiple joints; the very morphology that allows highly jointed structures to form smooth curves. To compound upon this complexity, tail tendons frequently show a branching morphology that manifests as multiple tendons arising from a shared muscle-tendon transition zone, tendons splitting distally into multiple branches before reaching their insertion sites on multiple vertebrae, and/or tendons forking near their insertion site to blend into the muscle-tendon transition zone of a small local intrinsic muscle slip [44–46]. This branching phenomenon greatly challenges musculoskeletal modeling processes that were developed for studying limb mechanics.

When designing software to digitally represent morphology, simplifying assumptions can inadvertently become restrictive constraints. This is evident in the modeling of complex muscle-tendon branching architectures. Due to the limitation of representing only one tendon per muscle, researchers typically adopt an approach utilizing multiple muscle-tendon units (MTUs) in parallel, each representing a unique insertion and a portion of the muscle associated with that insertion. For example, the extensor digitorum communis (EDC) muscle in humans has a tendon that divides into four distinct branches, each of which inserts into a different finger [47]. In existing musculoskeletal models of the human hand [8, 48, 49], the EDC is represented as multiple independent compartments, each associated with a specific tendon that extends to a particular finger. This modeling simplifies the representation of the muscle and tendon architecture. However, it may not fully capture the physical interactions that occur between the muscle and tendons, increasing the likelihood that such models may provide limited, or even misleading predictions.

Physical interactions between branched MTUs are likely to be particularly important in cases where the branches insert on different bones, the MTU spans many joints, or the MTUs form a complex branching network with more than two insertions. For example, consider an MTU that originates on one bone and then inserts on two different bones. If external forces bend the more distal joint, that branch of the tendon experiences stress and strain, which

increases stress on the tendon and muscle proximal to the branching site, but reduces stress on the parallel distal branch. This type of complex stress field has been essential to predicting tears in the human supraspinatus and infraspinatus [50, 51], but there are currently no examples of OpenSim models that reflect these empirical data. Here, we introduce a new modeling framework to explicitly model the mechanics of branched MTUs and use simplified case studies to estimate the effect of modeling strategy on motion predictions.

Current tools designed for musculoskeletal modeling are not well suited for handling the aforementioned modeling complexities for highly jointed systems. NMSBuilder, an open-source software to create subject-specific musculoskeletal models, was established to facilitate the model building process for lower limbs [52]. The resulting models can be easily exported to OpenSim. While NMSBuilder improves musculoskeletal modeling efficiency, the modeling flow still requires manual addition of every entity and is, therefore, not well-suited for modeling highly articulated systems, particularly those featuring complex muscle-tendon branching architectures. A similar challenge exists with OpenSim Creator [53], a newly introduced software still in its alpha stage, designed for creating musculoskeletal models. Furthermore, NMSBuilder only allows the use of a subset of modeling components of OpenSim; any elements absent in NMSBuilder, such as constraints and controllers, must be added manually post hoc in OpenSim if required. Other modeling software options, such as SIMM [54, 55], Anybody [56], and Visual 3D (Visual 3D Professional, C-motion, Germantown, MD, USA), are proprietary, expensive ($4995–7950 per year, with academic discounts applied), and tailored for limb modeling. Because none of these software packages provide full access to source code, it is difficult to expand their functionality to accommodate the complex tendon branching phenomena. Here, we introduce `ArborSim` (Articulated, Branching, OpenSim Routing), an extensible open-source tool integrated with the application programming interface (API) of OpenSim for constructing detailed musculoskeletal models of highly jointed systems, while accounting for complex muscle-tendon architectures. We intend the toolkit to enable biomechanics researchers to efficiently construct musculoskeletal models of highly jointed structures from CT scans and dissection data. We provide templates to store all essential data for modeling as Comma-Separated Values (CSV) files that are then used to automatically construct the model. This highly customizable framework enables users to easily set or update variables based on empirical data for any musculoskeletal parameter, even in highly complex systems.

In addition, we propose a novel approach to modeling complex muscle-tendon architectures that explicitly considers the interactions between each component. The approach is seamlessly integrated into the tool. By changing one line of code, users can decide to construct musculoskeletal models using either the conventional (parallel MTU) approach to modeling branching or our proposed method, both using the same data. We employ our tool to investigate the impact of using different methods to model MTU branching by constructing simplified, multi-jointed systems, i.e. "toy models." Large differences between the kinematic outputs of simulations using the conventional branching modeling method and the proposed method highlight the need for more accurate models of complex branching phenomena.

## Design and implementation

### Overview of `ArborSim`

A fundamental representation of a musculoskeletal system in OpenSim involves rigid bodies, joints, and MTUs. The rigid bodies represent bones and embody associated inertial characteristics including mass, center of mass, and moment of inertia that incorporate the mass of the muscles and tendons that move in tandem with each bone. These rigid bodies are interconnected through joints defined by Joint Coordinate Systems (JCSs). MTUs, functioning as

actuators, are represented by lines of action along with a set of parameters that govern muscle contraction dynamics. These lines of action are determined by MTU paths, which consist of user-selected path points. Additionally, wrap objects can be introduced to shape the line of action of an MTU along parametric surfaces, commonly known as wrapping surfaces.

The development of our tool centers around the procedural creation of these fundamental entities. To enhance the software's structure, reusability, and ease of maintenance, the tool is built using a four-layer architecture (see Fig 1). Among the four layers, the first and second layers (Fig 1A and 1B) serve as input interfaces, allowing users to input customized data for constructing an articulated musculoskeletal model of a highly jointed system. In these two layers, the first requires data in a "global" format, whereas the second needs "local" format data (More details on these formats are provided in subsequent sections). Users have the flexibility to input data in either format. If "global" data is provided, the "**Transformer**" feature will convert it into "local" format, ready for model construction.

The components of the musculoskeletal model, derived from the inputs of the first two layers, are then modified via the implementation of the proposed branching modeling approach, named "**Brancher**". These modified components reside in the third layer (Fig 1C). During this modification process, branched muscles and tendons—initially represented as multiple independent MTUs according to the conventional branching modeling method—are altered to conform with the proposed branching modeling method. Subsequently, depending on the desired branching representation, the tool assembles the stored components either in the second or third layer via the functionality named "**Builder**", thereby generating and exporting the resulting musculoskeletal model that can be directly loaded into OpenSim. This resulting model constitutes the fourth layer (Fig 1D).

Each category of data within the first two layers is stored in a dedicated CSV file. The natural data structure and ubiquity of CSV files offer a clear and intuitive way to represent the data, and simplify data sharing and collaboration with researchers using diverse tools or platforms. Moreover, the tool can be easily extended, as one can create additional dedicated CSV files to accommodate extra components and functionalities for the tool.

## Layer 1

In the first layer, users provide the following categories of data:

- Coordinates of anatomical landmarks for establishing body frames.

- Coordinates of anatomical landmarks for defining JCSs.

- Coordinates of anatomical landmarks outlining paths for MTUs.

- Coordinates of anatomical landmarks for wrapping surfaces.

- Physiological parameters of MTUs.

- Segmented bone geometries.

- Branching groups.

- Joint range of motion.

Among the data mentioned above, the coordinates of anatomical landmarks are represented in a global reference frame. In the context of working with a CT scan or other biomedical imaging resources, these landmarks can be identified in segmentation and visualization software such as Mimics (Materialise NV, Leuven, Belgium), 3D Slicer [57], and ITK-SNAP [58]. In this context, the term "global" coordinates of these landmarks refers to the coordinates

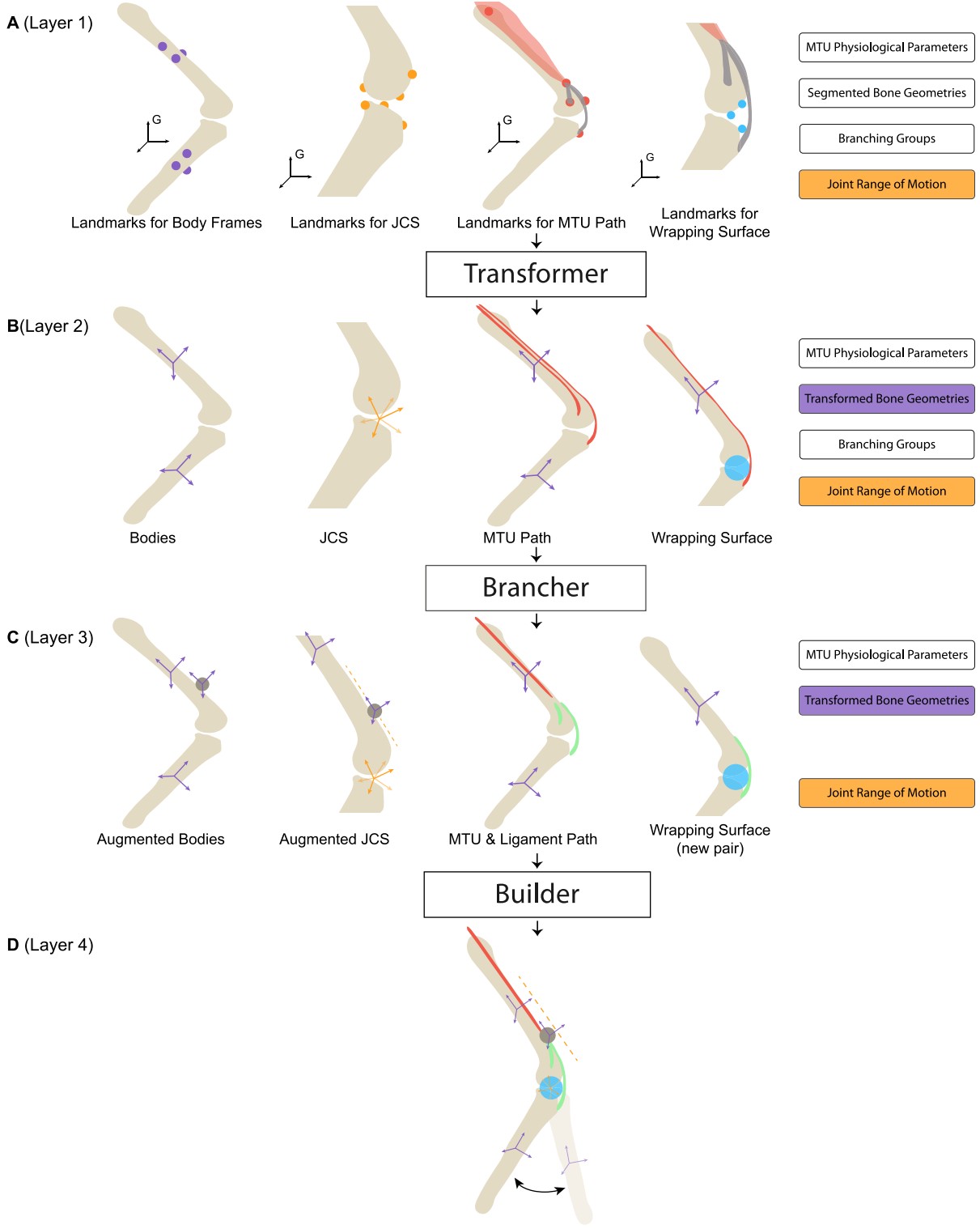

**Fig 1. Overview of ArborSim.** (**A**) The input layer with global data, divided into eight categories. In the diagrams, purple dots indicate anatomical landmarks used for establishing body frames. Orange dots represent landmarks for defining JCSs, while red dots denote landmarks for MTU paths. Blue dots are used to mark landmarks for defining wrapping surfaces. Transparent red contours depict the muscle, grey slender contours outline the tendons, and the black frames labeled "G" represent the reference frame in which the landmark coordinates are collected. (**B**) The input layer with local data. Separate MTUs (red) are used to represent the muscle-tendon structure. (**C**) The middle

layer with modified components following the proposed branching modeling method. The dark grey circles denote the extra body introduced to connect the single MTU (red) and the passive elements (green). (**D**) The output layer with the constructed musculoskeletal model.

within the default coordinate system in which the CT scan is saved and visualized. This default coordinate system provides a common and consistent spatial reference for all the data in the CT scan.

To establish a reference coordinate system within the current software setup, it is required to provide three landmark points: one for the origin of the coordinate system, one to denote the positive X-axis direction, and another to denote the positive Y-axis direction (Fig 2). Thus, for each body, users provide coordinates of three anatomical landmarks for establishing the corresponding body frame.

To create a joint that defines the kinematic relationship between two bodies, termed parent and child bodies, two joint frames are constructed, each affixed to the parent body and child body (see Fig 3), respectively. Therefore, six landmark points are required to construct the joint frames for each joint. Three of these landmarks specify the location and orientation of the joint frame on the parent body, and the other three specify the location and orientation of the joint frame on the child body. During the JCS landmark identification process, some recommended guidelines for standardized JCS definitions can be found in [59–61].

To maintain compatibility with the conventional method of modeling branched muscles and tendons in cases of complex branching phenomena, and to seamlessly incorporate the proposed branching modeling method, offering flexibility in creating models using either technique, the MTU-related data provided in the input are stored in a manner that presents branched muscles and tendons as distinct, separate, and independent muscle-tendon compartments. This adheres to the rule of a single tendon per muscle in OpenSim. For instance, when a muscle is attached to a tendon that branches into two insertion points (see Fig 1), two MTUs are used to represent this muscle-tendon structure. The anatomical landmarks for MTU path are provided for each MTU, featuring identical landmarks along the muscle pathway, and different landmarks along the tendon pathway. Correspondingly, two sets of physiological parameters are designated for these two MTUs, encompassing divided maximal isometric force, consistent pennation angle and optimal fiber length, and different tendon slack length, etc..

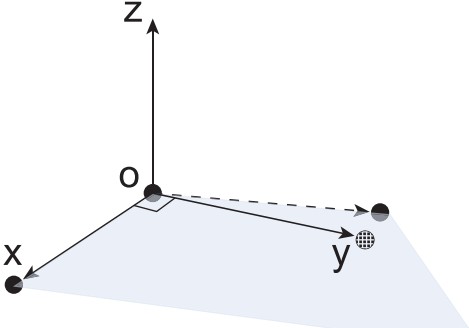

**Fig 2. Coordinate system construction.** Three solid black dots indicate the landmarks for the origin and the positive directions of the X and Y axes, respectively. The plane, depicted in light blue, is defined by these points. The dashed black dot represents the adjusted landmark for the positive Y-axis direction, ensuring orthogonality with the X-axis.

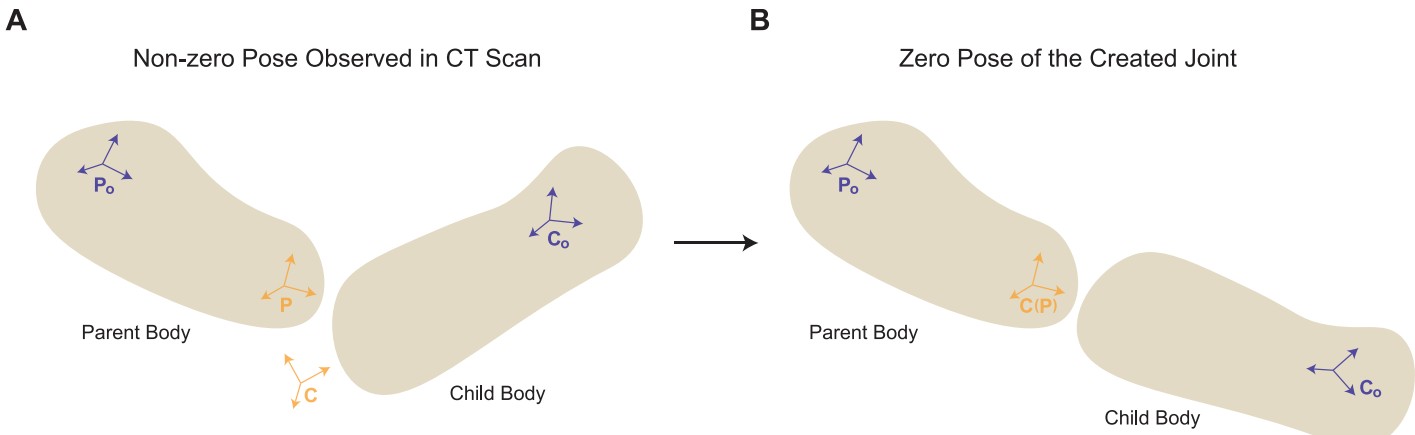

**Fig 3. Joint construction.** A joint that defines the kinematic relationship between frames P and C is created, where frame P is affixed to the parent body, with its body frame denoted by $P_o$, and frame C is affixed to the child body, with its body frame denoted by $C_o$. By making frames P and C coincident, the non-zero pose observed when collecting landmarks to build the JCS from CT scans or other data sources, shown in (**A**), is transformed into the default zero-pose, shown in (**B**).

To enable MTUs to interact with certain wrapping objects, mimicking the way muscles and tendons navigate around bones and other structures, wrapping surfaces can be established. These surfaces guide the paths of linked MTUs in a manner that cannot be achieved with fixed MTU path points alone. To create a wrapping surface, provide three anatomical landmarks that define the reference frame, as well as additional geometry parameters such as the radius and height of the wrapping surface.

To prepare for implementing the proposed branching method, i.e., the conversion from the second layer to the third layer, the fact that multiple MTUs together form a branching unit within the biological equivalent is preserved as branching group data.

For other essential data, the segmented bone geometries enable effective visualization within OpenSim. Notably, these segmented geometries are exported directly from the same global reference frame used for collecting landmark coordinate data, aligning with the default frame of the CT scan or other imaging data.

The joint range of motion data restricts the relative movement of the bodies. Such data might not be immediately accessible to users until the joints are precisely defined in the model as this information is more readily obtainable in local frames as opposed to a single global frame. Therefore, users may initially overlook this aspect and later assign joint range of motions after constructing a model devoid of embedded joint range of motion. In a model, joint range of motion can be determined by ensuring no penetration occurs between the parent and child body of a joint. Experimental data can also be instrumental in this process.

## Transformer creates layer 2

In contrast to data in the first layer, which is collected under the same global coordinate system, OpenSim models use distinct reference frames for each body, making the model more biomechanically relevant and intuitive. Consequently, it necessitates the transformation of each segmented bone geometry, initially exported from the segmented CT scan, so that each body in the resulting model has its own local coordinate system. Additionally, elements such as joints, MTU paths, wrapping surfaces, etc., are constructed based on the relationships between associated bodies. This construction requires transforming their "global" coordinates into the respective "local" coordinate systems of the associated bodies.

Within the tool, when the "global" data and other necessary data are input into the first layer, the tool seamlessly performs the required transformations from the "global" data to a "local" format, while keeping other necessary data unchanged. From there, users can obtain bodies, JCSs, MTUs, and wrapping surfaces that are directly compatible with OpenSim, all of which are stored in the second layer. This conversion is particularly advantageous as it eliminates the need for manual transformations by users. Consequently, users can directly integrate data obtained from other software tools, thus streamlining the model development process.

A detailed explanation of the automated transformation from "global" to "local" is provided as follows.

As stated in the previous subsection, to establish a reference coordinate system, three landmark points are provided. These three landmark points combine to form two vectors, representing the reference X and Y-axes (see Fig 2), respectively.

When identifying these landmarks, it is very likely that these two vectors are not perfectly perpendicular to each other due to various factors, such as measurement errors during the process. This does not conform to standard Cartesian geometry. To address this, the tool adjusts the location of the landmark indicating the positive direction of the Y-axis. This is achieved by locating a new point on the plane defined by these two vectors that is closest to the original landmark for the positive Y-axis direction, and ensuring that the new point creates a resulting angle of 90 degrees between the reference X-axis and corrected reference Y-axis. Subsequently, the reference Z-axis can be defined by taking the cross product of the X-axis vector and corrected Y-axis vector. Finally, we normalize these vectors by dividing each by their respective magnitudes to establish a well-defined coordinate system.

By following the procedure outlined above, body frames are established using the "global" coordinates of three anatomical landmarks for each body from the supplied data in the first layer of the tool. From the normalized vectors, which represent body frame axes, we can build transformation matrices. These transformation matrices can then transform the original segmented bone geometries from the "global" coordinate system to their individual "local" coordinate systems. As a result, with the transformed bone geometries and associated center of mass and inertia defined in the corresponding body frames, a set of rigid bodies—a primary building block of the model—can be readily derived using OpenSim's API.

With six anatomical landmarks provided for each joint construction, as explained earlier, three represent the joint frame's location and orientation on the parent body, and the other three represent those on the child body. Following the method used to transform bone geometries from "global" to "local", a constant transformation that represents the offset between a joint frame and the corresponding body frame can be calculated. That establishes the JCSs.

Joints are modeled as CustomJoints, which is the most generic joint representation in OpenSim. It allows for motion in 6 DOFs, offering the flexibility in modeling traditional joints (e.g., slider joint, universal joint) as well as complex biomechanical joints.

Implementing the same transformation concept, the positions of MTU path points relative to the body frames through which the MTU traverses can be calculated. Similarly, the location and orientation of a wrapping object in relation to the associated body segments can also be determined.

The aforementioned section explains how the tool efficiently converts the provided "global" data into a "local" format, designed to integrate smoothly with OpenSim. Alternatively, if users opt to directly provide local data and construct a model from there, they can bypass the first layer and simply supply the local data through the second layer. This feature allows users to create artificial toy musculoskeletal models conveniently, as working with local data aligns more intuitively with the understanding of the desired local relationships.

## Brancher creates layer 3

As briefly touched on in previous sections, the traditional representation of a muscle-tendon branching structure involves multiple independent MTUs (see Fig 1B). In that representative scenario, two distinct MTUs are used to account for the two tendons with different insertions. Correspondingly, the muscle is represented as two compartments, each associated with a tendon. Both MTUs stem from the same muscle origin, following the same path up to the branching point, then diverging along their unique tendon paths.

The physiological parameters of the two MTUs depend on the respective muscle compartment and tendon. Usually, they share the same muscle pennation angle, and possess nearly identical optimal fiber lengths. The maximal isometric force of the muscle is allocated based on the physiological cross-sectional area (PCSA) of each muscle compartment, and the tendon slack length of each MTU aligns with the tendon's resting length. Collectively, these parameters characterize the active and passive force-length curve, the force-velocity curve of the muscle, and the force-length curve of the tendons, determining the force development of MTUs in a musculoskeletal model. To synchronize the functioning of these muscle compartments, two identical muscle excitation signals can be applied across the MTUs.

In contrast to the conventional method, our proposed branching modeling approach utilizes a single MTU to represent the segment of muscles and tendons that share the same pathway from their origin, prior to the emergence of any branching, within a group of original MTUs specified in the provided branching group data. Hence, in the representative scenario (Fig 1C), the muscle is no longer fragmented into two compartments. Instead, a single MTU is used to represent the muscle in the real physiological system. The maximal isometric force of this single MTU equals the sum of the maximal isometric forces from the two MTUs in the conventional branching modeling method, while its optimal fiber length and pennation angle remain consistent with those of the two MTUs.

There are two cases of tendon branching that must be modeled slightly differently. The muscle-tendon unit depicted in Fig 1C directly branches into two tendons at the muscle-tendon transition. In this case, the single MTU would ideally represent a muscle without tendons. However, because OpenSim does not support MTUs with zero tendon (slack) length, an exceedingly short tendon can be introduced into that single MTU. This ensures that the MTU predominantly represents the muscle. To prevent the stretching of this extra tendon from affecting muscle force distribution and coordination to the downstream tendons, the short tendon can be made rigid. Alternatively, when a tendon segment is connected to the muscle before branching occurs, the additional rigid short tendon becomes unnecessary. Instead, we can simply use an MTU to represent the muscle associated with that particular tendon.

To represent the tendons post-branching, ligaments are used. In OpenSim, as in biomechanics in general, a ligament represents passive connective tissue that connects bones to other bones. Both the tendon in an MTU and a ligament are defined by a normalized force-length curve that describes how the force responds to being stretched, a force scale that scales the normalized force-length curve, and the resting length. Therefore, by assigning the same properties, they are treated equivalently in OpenSim.

To emulate the branching architecture, extra bodies, modeled as spheres, are introduced for each branching point. As branching points in reality can slide relative to vertebrae, a CustomJoint is incorporated between each extra body and its corresponding vertebra to ensure this movement capability. Within the CustomJoint, the three rotational DOFs are restricted, rendering it solely as a sliding joint. To effectively negate the impact of inertia changes induced by the inclusion of the extra body to the model on the force development and the overall motion, the mass of the extra body is set to an exceptionally low value. For the representative

muscle-tendon structure, one extra body is employed at the muscle-tendon transition point, accounting for the muscle dividing into two tendons. By linking the extra body to the respective tendon insertion points using ligaments, the muscle and tendons in the musculoskeletal system are inherently interconnected in the model. Therefore, the intricate interactions and dependencies within the musculoskeletal system can be more faithfully represented.

In the proposed branching modeling method, the resting length of each ligament should match that of its corresponding tendon segment in the actual system. The force scale for each ligament should be allocated as a portion of the muscle's maximal isometric force, because the force in the normalized force-length curve is normalized by the muscle's maximal isometric force in OpenSim. If the tendons have uniform material properties, the ligament's force scale can be determined by distributing the maximal isometric force based on the tendons' cross-sectional areas.

In summary, based on the data from the second layer, where the elements of the musculoskeletal model align with the traditional branching modeling approach, the proposed branching modeling method demands several adaptations. Specifically, these include the addition of supplemental bodies to the original model, the designation of sliding joints for these newly added bodies, adjustments to MTU pathway, the integration of ligaments to denote the branched tendons, recalibration of muscle-tendon physiological parameters, and updates to the paired association between the wrapping surfaces and the MTUs/ligaments. This conversion is inherently integrated into the tool, facilitating the creation of models using the proposed branching modeling method. The detailed procedure for the implementation of this conversion is expounded in Algorithm 1.

**Algorithm 1: Brancher**

```
Step 1. Construct Directed Tree Graph
foreach branching group do
  foreach MTU in the branching group do
    foreach path point of the MTU (from its origin to its insertion) do
      Check for existing vertex by associated body, the position on
      that body, and the point type
      if vertex exists then
        Append the path point information (MTU name, index, type) to
        the vertex
      else
        Create a new vertex for the path point
        Associate the path point information to the vertex
        if path point is not the origin then
          Create a directed edge from the previous path point vertex to
          the new vertex
Step 2. Identify Special Vertices
foreach vertex in the graph do
  if vertex's indegree × vertex's outdegree ≠0 and vertex's indegree
    + vertex's outdegree >= 3 then
      Mark the vertex as a branching vertex
  if vertex represents an MTU's origin then
      Mark the vertex as an origin vertex
  if vertex represents an MTU's insertion then
      Mark the vertex as an insertion vertex
Step 3. Add Extra Bodies and Sliding Joints
foreach branching vertex do
  Construct an extra body modeled as a sphere
  Build a CustomJoint with translational DOFs between the associated
  body of the vertex and the sphere
Step 4. Refactor MTU Paths and Build Ligament Paths
```

```
foreach origin vertex do
  Traverse the graph until reaching any branching vertex
  Model this segment as an MTU, originating from the origin vertex's
  associated body, inserting to the branching vertex's associated
  extra body
foreach branching vertex do
  Traverse the graph until reaching a branching or insertion vertex
  Model this segment as a ligament, originating from the branching
  vertex's associated body, inserting to the branching or insertion
  vertex's associated (extra) body
```

In the algorithm, directed tree graphs are constructed to represent the muscle-tendon branching structure. By analyzing these tree graphs, the algorithm automatically makes the necessary adjustments and outputs the modified components into CSV files, except for the reassessment of muscle-tendon physiological parameters. Users manually input physiological parameters for the modified MTUs and newly introduced ligaments in the corresponding CSV file, based on anatomical data.

## Builder creates layer 4

Leveraging OpenSim's API, the modified components in the third layer that fits into OpenSim are integrated via the functionality named "**Builder**" to create a musculoskeletal model. In this model, the branched muscles and tendons are modeled using the proposed method. As mentioned earlier, the components in the second layer present the branched muscles and tendons using the conventional method, and these components are also directly compatible with OpenSim. Therefore, to create a musculoskeletal model using the conventional branching modeling method, users can simply bypass the third layer and construct a model from the second layer. Thus, the tool offers the capability of creating two musculoskeletal models, using two different branching modeling methods, both utilizing the same data.

## Comparative simulation case study

Here, we employ ArborSim to investigate the tangible distinctions between the conventional parallel modeling method and the proposed branching modeling method for modeling complex muscle-tendon architectures. The investigation used comparative simulations conducted on artificial toy articulated musculoskeletal systems. Within these systems, we created muscle-tendon architectures with branching features. We focused on the branching of tendons rather than muscles, in line with our findings in mammalian tails [45, 46].

Drawing from observed variations in muscle-tendon branching architectures across a variety of mammalian tails [45, 46], we identified three relatively prevalent categories of variation, which serve as the main focus of our comparative simulations:

- Variations in the number of tendon branches into which a muscle divides.

- Variations in the number of joints between the terminals of the branched tendons.

- Variations in the ratio of muscle fiber length to MTU length.

These categories are illustrated in Fig 4. For each category, six musculoskeletal systems were designed. To simplify the comparison between the traditional branching modeling approach and our proposed method, we restricted these toy musculoskeletal models to a planar configuration. Each toy musculoskeletal model consisted of 8 vertebrae connected by planar joints, with each vertebra modeled as a uniformly dense cylinder in OpenSim. The cylinders had a mass of 0.1 kg, length of 0.2 m, and radius of 0.025 m. A gap of 0.025 m

**A**  Tendon Branch Count    **B**  Joint Count Between Insertions    **C**  Muscle Fiber Length / MTU Length

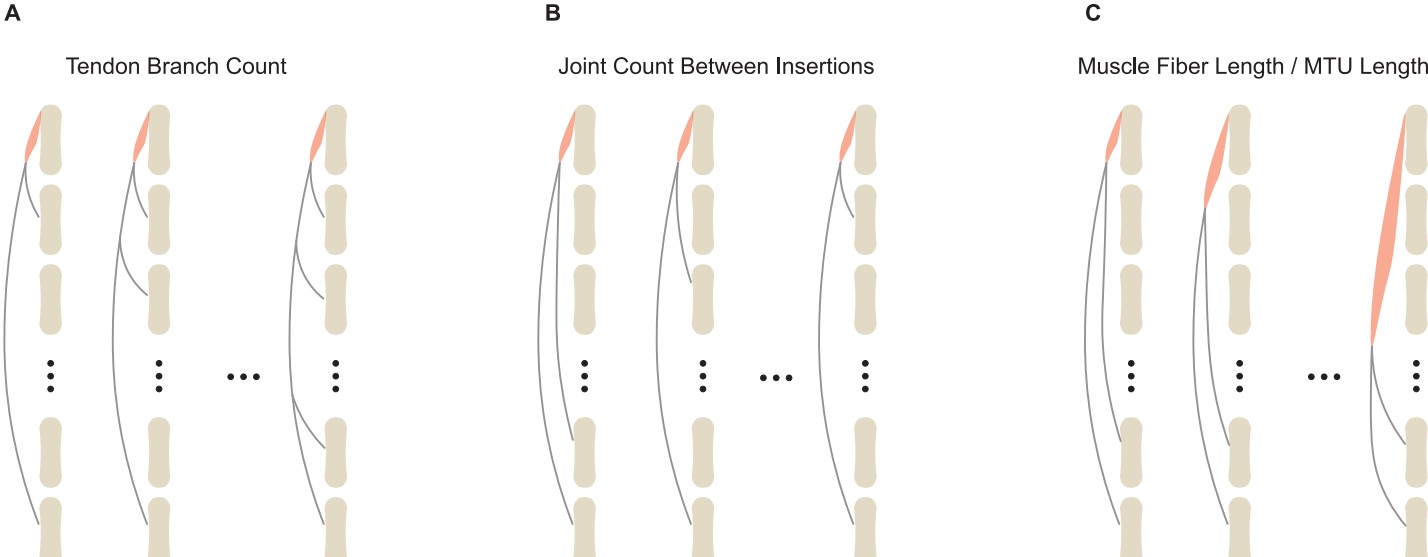

**Fig 4. Comparative simulation categories.** The category of varying numbers of tendon branches is shown in (**A**). The category representing the varying numbers of joints spanned between insertions is depicted in (**B**). The category showing the varied muscle fiber length to the longest MTU ratio is presented in (**C**). In all three diagrams illustrating these categories, the red contours indicate the muscles, while the grey slender contours represent the branched tendons. To more clearly visualize the branching, the tendons and muscles in the diagrams are displayed away from the bones. However, in the models, they are all connected to the bones through the use of "via points" on the bone surfaces, allowing them to exert force on the bodies along the pathway.

between each pair of consecutive vertebrae was introduced to prevent interpenetration during movement. The range of motion for each joint was calculated based on the cylinder dimensions and the gap length, with CoordinateLimitForce in OpenSim applied to enforce these constraints.

Each system was equipped with a muscle-tendon branching architecture, tailored to its respective category. Each muscle-tendon branching architecture consisted of a single muscle and multiple branched tendons. To ensure that all vertebrae actively respond to muscle activation, a long tendon spanning from the muscle-tendon transition area to the most distal vertebra (bottom in Fig 4) was present in all the systems. In the category where the ratio of muscle fiber length to MTU length varied (Fig 4C), the MTU length was denoted by the measurement of the longest MTU in a given model.

For each system, two musculoskeletal models were constructed: one using the conventional branching modeling method and the other using the proposed branching modeling method, following the steps presented in previous subsections (note that both branch modeling methods are incorporated within ArborSim). Given the physiological parameters of the muscle, for models using the conventional branching modeling method, the maximal isometric force of the muscle was evenly allocated among these MTUs. For models using the proposed branching modeling method—where a single MTU is used to represent the muscle—the maximal isometric force was directly assigned to that MTU. For fair comparison between equivalent models, we divided the total maximum isometric force of the muscle by the number of insertions to set the force scale for each distal ligament element. The force scale for the proximal ligaments was then set to the sum of the force scales of the connected distal ligaments.

Regarding other MTU and ligament modeling parameters, the optimal and default muscle fiber lengths were defined as the distance between the muscle's origin and insertion in a zero-pose, where all the joint angles are zero, resulting in a straight, fully extended pose. Similarly,

the tendon slack lengths for MTUs in the conventional method, as well as the resting lengths for ligaments in the proposed method, were established based on the default lengths of their respective segments in this zero-pose configuration. For the branched model, an extremely short rigid tendon of 0.0001 m was included in the MTU that represents the muscle to circumvent OpenSim's limitation of not supporting MTUs with zero tendon length.

To ensure comparability between the conventional and proposed methods, the same normalized force-length curves and force-velocity curves were used for the muscle. Additionally, the normalized force-length curve of the tendon in the MTU was identical to that of the ligament. The muscle was modeled as a Millard2012EquilibriumMuscle [62] in OpenSim. The mass of the extra body introduced when using the proposed branching modeling method was set to $5 \times 10^{-4}$ kg.

Within each category, an identical muscle excitation signal was applied to all systems with varying muscle-tendon architectures, and simulations were run under that excitation signal for a duration of $T = 3$ seconds. To evaluate the difference between the traditional branching modeling method and proposed method on the resulting kinematics, we defined the following metric:

$$\Delta = \max_t \sum_{i=1}^{n} \frac{|\theta_i^{con}(t) - \theta_i^{pro}(t)|}{\bar{\theta}_i} \times 100\%, \qquad t \in [0, T] \tag{1}$$

where $\theta_i^{con}(t)$ denotes the joint angle of $i$-th joint at time $t$ for models generated using the conventional branching modeling method, $\theta_i^{pro}(t)$ denotes the joint angle of $i$-th joint at time $t$ for models generated using the proposed branching modeling method, $\bar{\theta}_i$ denotes the positive bound of $i$-th joint, and $n$ is the number of joints, which is 8 in our study. In the metric, the numerator $\sum_{i=1}^{n} |\theta_i^{con}(t) - \theta_i^{pro}(t)|$ captures the total absolute differences in joints angles across all eight joints at time $t$. The denominator $\sum_{i=1}^{n} \bar{\theta}_i$ represents the sum of the positive bound of each joint. This reflects the pose where the toy musculoskeletal system is rotated to its limit.

Therefore, the metric quantifies the maximal percentage of the total absolute differences in joint angles across all eight joints over the entire motion trajectory, relative to the joint limits. This metric highlights the worst-case scenario in terms of the discrepancy between the two methods, providing a robust evaluation of their impact on system behavior.

## Sensitivity analysis

Performing a sensitivity analysis is valuable for assessing the impact of potential measurement errors on the parameters of a model, providing insights into the model's robustness. Results from previous extensive sensitivity analyses, particularly focusing on the effects of Hill-type muscle-tendon model parameters within musculoskeletal models [63–70] indicate that the tendon slack length and maximal isometric force are the most sensitive parameters. Therefore, in this study, we conducted a sensitivity analysis on our proposed branching modeling method, focusing on these two parameters, to investigate their influence on the joint kinematics of the toy musculoskeletal models introduced in the previous section. Because all the tendons in these toy musculoskeletal systems were represented by ligaments, the sensitivity analysis on the tendon slack length should be equivalent with a sensitivity analysis on the ligament resting length in this context.

To assess the influence of changes in maximal isometric force, we applied ten perturbations to their nominal values, varying from −10% to + 10% in 2% increments. Initially, we planned to use this same range for the nominal ligament resting length in each toy model. However, in most toy musculoskeletal systems, tendons are considerably longer compared to muscle length. Consequently, substantial perturbations, particularly negative ones, to ligament resting

length could lead to simulation failures in OpenSim. This issue arises because negative perturbations cause the ligament to contract at the start of the simulation when the muscle is not fully activated. The larger the negative perturbation, the more the ligament contracts. If such contraction moves the extra body, representing tendon branching points outside the two consecutive via points at the ends of the cylinder body, the simulation halts. As a result, we reduced the perturbation range for ligament resting length to −5% to + 5%, also in 2% steps. Notably, even this altered range is large due to the tendons' considerable length in these systems. While it was possible to address simulation crashes by manually adjusting parameters in the toy models, we avoided this due to the numerous parameters, their complex interrelations, and their complex effects on the resulting motions, especially in the context of branching. Therefore, we chose to simply reduce the perturbation range.

It is important to note that as there was more than one tendon in all of these toy musculoskeletal systems; we have chosen to apply the perturbation of the ligament resting length to all the ligaments simultaneously. For each perturbation, a new simulation was run under the same excitation signal used for the original experiments. Sensitivity analyses were conducted on both the conventional method and the proposed method.

To quantify the effect of the perturbation on the resulting motion, we re-applied the metric introduced:

$$\Delta = \max_t \sum_{i=1}^{n} \frac{\left| \tilde{\theta}_i^{pro}(t) - \theta_i^{pro}(t) \right|}{\bar{\theta}_i} \times 100\%, \qquad t \in [0, T] \tag{2}$$

This time, instead of comparing the proposed method with the conventional method, we compared the perturbed with unperturbed model for both the proposed method and the conventional branching modeling methods. Here, $\tilde{\theta}_i^{pro}(t)$ is the perturbed joint angle of $i$-th joint at time $t$. $\theta_i^{pro}(t)$ is the unperturbed joint angle of $i$-th joint at time $t$.

## Results and discussion

### Model construction

To demonstrate the use of the designed software for building articulated musculoskeletal models of highly jointed systems, the toy musculoskeletal systems introduced in the previous section were constructed by running through the entire software model construction pipeline. In addition, joint coordinate reporters and muscle controllers are assigned to each model using the OpenSim API. This setup allows for the recording and post-analysis of simulated motion in response to predefined muscle excitation signals.

The CSV data files and code for this model construction and the associated simulation results are available at https://github.com/EMBiRLab/ArborSim.

### Comparative simulations

The joint kinematics of the musculoskeletal models, which represent articulated systems built using the conventional and proposed methods, showed large differences. The quantitative differences between the derived motions, evaluated using metric (1), are presented in Fig 5.

Specifically, within the category of different numbers of tendon branches (Fig 5A), the range of $\Delta$ across six simulations was [26.33%, 66.31%]. The peak value of this difference occurred when there were five tendon branches. Overall, as the number of tendon branches increased, the difference in joint kinematics first decreased and then increased. This trend suggested a complex, nonlinear relationship in between branched tendon systems.

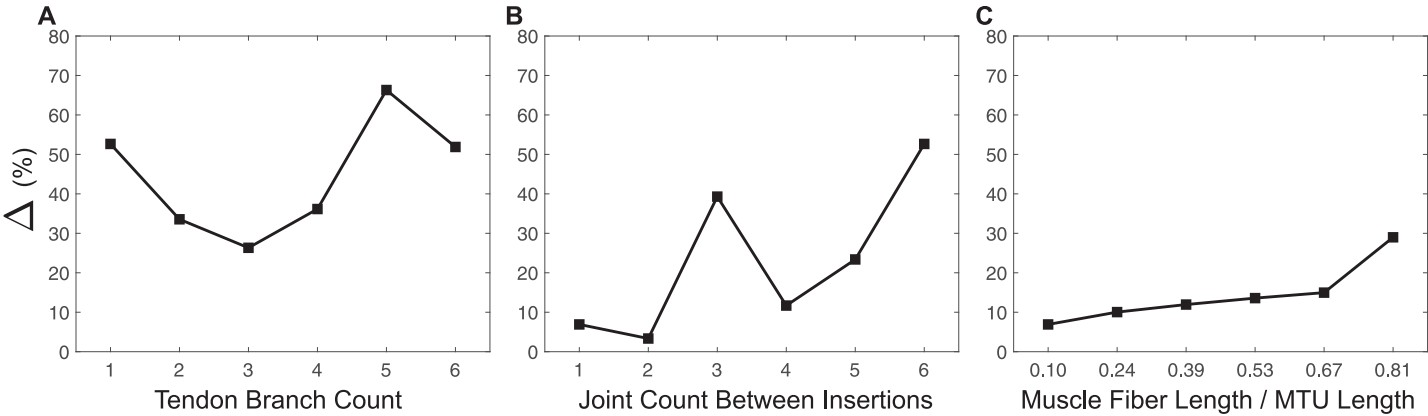

**Fig 5. Quantitative motion comparison.** Quantitative differences between the proposed branching modeling method and the conventional branching modeling method, evaluated by metric (1), on the resulting motions of articulated systems with a varied number of tendon branches (**A**), a varied number of joints between insertions (**B**), and varied muscle fiber length to the MTU length ratios (**C**) are shown, respectively. Note that the ratios of muscle fiber length to MTU length presented in (**C**) correspond to those ratios at zero-pose.

Within the category of different numbers of joints between insertions (Fig 5B), the range of $\Delta$ across six simulations was [3.34%, 52.65%]. The largest $\Delta$ occurred when the branched tendons in the models spanned six joints between insertions. Disregarding the abrupt increase in the percentage change at three joint counts between insertions, a general incremental trend was observed in the motion differences with respect to the joint count between insertions. However, this jump might once again indicate a more complex, possibly nonlinear relationship.

Within the category of different muscle fiber length to MTU length ratios (Fig 5C), the range of $\Delta$ across six simulations was [6.91%, 28.99%]. The largest $\Delta$ occurred when the muscle spanned five joints, with the corresponding ratio being 0.81. There was a consistent trend of increasing motion differences as the muscle fiber length to MTU length ratio increased. However, compared to the other two categories, variations in the muscle fiber length to MTU length ratios resulted in relatively smaller changes and variance in $\Delta$ values.

To visually demonstrate the motion differences between the two modeling methods, we included a figure that highlights the comparison with the largest kinematic differences, determined by the metric (1), across each category (Fig 6). Animated motion comparisons throughout complete simulations are provided in S1–S3 Videos files.

Collectively, these results indicate that the conventional and proposed branching modeling methods exhibit notable differences in predicting joint kinematics. In particular, these differences become more pronounced as the complexity of muscle-tendon branching architectures increases, which involves numerous tendon branches, and an extensive number of joints spanned by branched tendon insertions. The resulting large differences can be concerning in fields demanding precise movement prediction, such as surgery outcomes, especially as the complexity of the muscle-tendon networks escalates in systems like human hands, spines, and elongated tails.

To delve further into the source of the differences between the proposed method and the conventional method, we examined changes in muscle fiber length over time for each method. Because muscle fiber length affects contraction velocity and force, it is important to correctly model changes in muscle fiber length through time. In a system with one muscle attaching to multiple tendons, an accurate MTU model should reflect the uniform muscle fiber length changes within that single muscle. Using the conventional method with independent MTUs

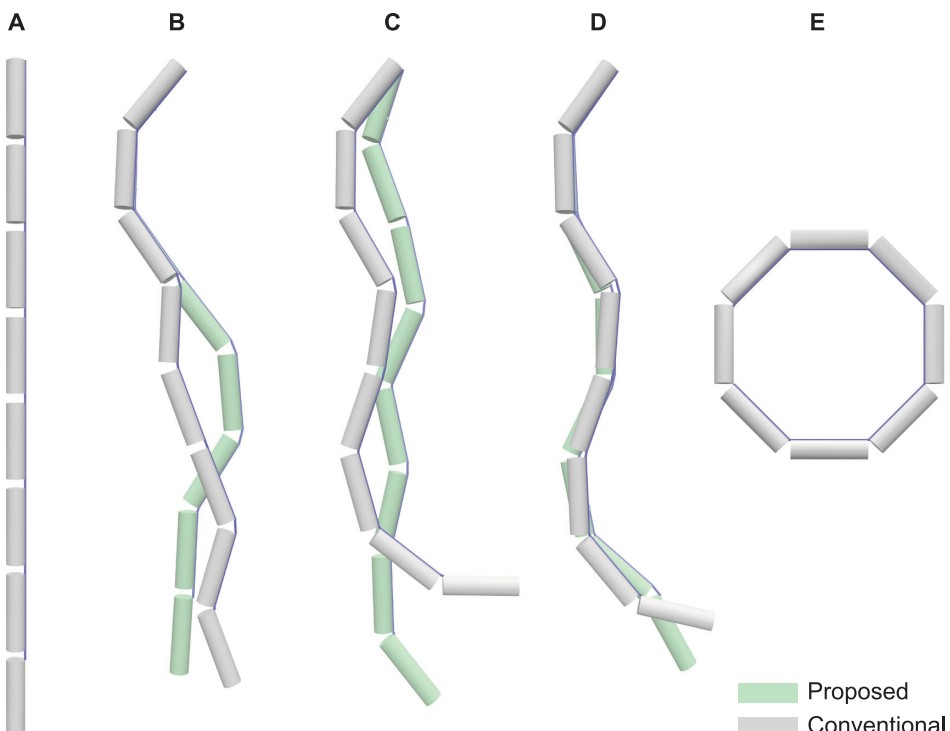

**Fig 6. Qualitative motion comparison.** (**A**) displays the zero-pose of the models. For (**B**), (**C**), and (**D**), each panel presents two comparative snapshots from the simulation that exhibited the highest Δ out of six simulations per category. These snapshots represent the largest motion differences between the conventional branching modeling method (depicted in white grey) and the proposed branching modeling method (depicted in light green). Specifically, (**B**) compares models with varying numbers of tendon branches, (**C**) compares models with different joint counts between insertions, and (**D**) comprares models with varied muscle fiber to MTU length ratios. (**E**) illustrates the pose where each joint is set to its maximum value within the range of motion.

reflecting each tendon insertion point, the normalized fiber length changes over time are not uniform (Fig 7, gray lines). This discrepancy indicates that the conventional branching modeling method fails to explicitly consider the physical interactions between the entities in a complex muscle-tendon network. In contrast, the proposed branched modeling method only includes one MTU, inherently constraining all muscle fiber length changes to be uniform, as in a real muscle (Fig 7, green line).

The results here highlight the role of the physical interactions between branched muscles and tendons in highly jointed systems. The study, though preliminary and focused on a kinematics comparison in toy musculoskeletal systems, underscores the necessity for continued in-depth exploration and advancement in the understanding of complex muscle-tendon branching phenomena and the branching modeling techniques in future works.

In this study, we employed toy musculoskeletal models to compare the conventional branching modeling method with our newly proposed approach. Future implementation of our methods will involve validation with real *in vivo* data.

Validation can be performed at two different hierarchical scales. One is solely on the scale of the branched muscle-tendon network. This can be pursued by collecting data on contraction, the distributed interaction forces of muscles and tendons, and other relevant factors when a load is applied to a branched muscle-tendon network. One can follow an approach similar to that outlined in [51], which examined the strain in branched tendons. To align

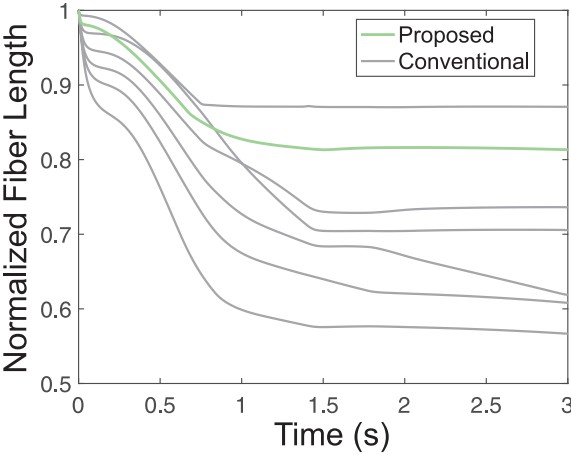

**Fig 7. Normalized muscle fiber length comparison.** The plot compares the normalized muscle fiber lengths of the independent MTUs modeled using the conventional method with those of a single MTU modeled using the proposed branching method. Here, "normalized" refers to the normalization of fiber length relative to the optimal fiber length. The comparison highlights the case with the largest Δ, as shown in Fig 5, corresponding to a Tendon Branch Count of 5 in Fig 5A. In the plot, normalized muscle fiber lengths are represented in light gray for the conventional method and light green for the proposed method.

experiments with simulations, we need detailed characteristics of each individual muscle and tendon within the network, including maximal muscle force, default fiber length, and tendon slack length for each branch. By measuring these specific parameters, we can replicate the experiments under identical loading conditions, thus allowing a direct comparison of experimental and simulation data to validate our branching modeling method.

Another hierarchy of validation would involve the integration of the branched muscle-tendon network with skeletal bodies, and the comparison of the resulting motions with both the traditional and proposed branching modeling methods. Such validation would require the construction of detailed musculoskeletal models with branched tendons, such as mammal tails [45, 46]. To achieve this, in addition to the muscle-tendon-wise parameters, capturing muscle excitation and joint kinematic data would be helpful.

## Sensitivity analysis

The resulting deviations from the nominal movement, evaluated by (2), showed large variability across different categorized muscle-tendon branching structures. The effects of the perturbations on maximal isometric force, and ligament resting length on the joint kinematics are shown in Figs 8 and 9, respectively.

Regardless of the muscle-tendon branching structure category, the models' joint kinematics was more sensitive to ligament resting length than to muscle maximal isometric force. This is consistent with previous studies [63–68], emphasizing the crucial role of the tendon slack length plays in the joint kinematics of the musculoskeletal systems.

Regarding the sensitivity to muscle maximal isometric force, specifically within the categories of varying numbers of joints between insertions (Fig 8B) and varying muscle fiber length to MTU length ratios (Fig 8C), the observed Δ values were consistently low and stable. This suggests a low impact of these categories and variations within these categories on joint movement sensitivity. Within the category of varying numbers of tendon branches (Fig 8A), systems with fewer than 4 tendon branches exhibited a similar low sensitivity. However, for systems

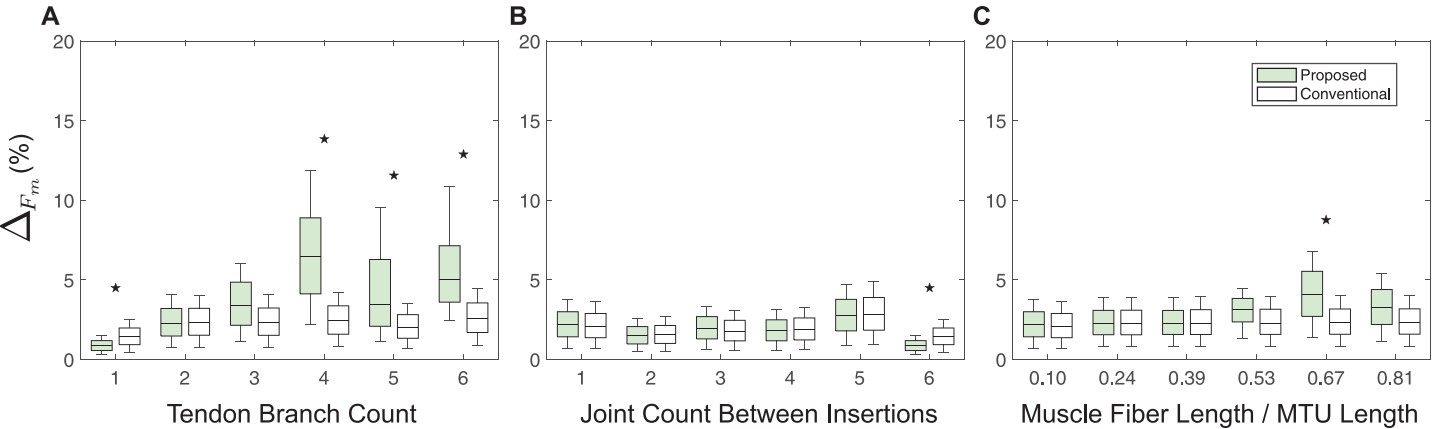

**Fig 8. Maximal isometric force sensitivity analysis.** Impact of maximal isometric force changes in the proposed versus conventional method on articulated system movement, measured by metric (2) across variations in tendon branch count (**A**), joint count between insertions (**B**), and muscle fiber length to the MTU length ratio (**C**). Solid black stars, if present, above the box plots indicate that the difference in Δ values between two groups, modeled by the conventional and proposed methods, is significant, determined by a two-tailed T-test (P-value < 0.05). The sample size for each box in the box plot was 10, corresponding to ten perturbations of nominal values adjusted from −10% to + 10% in increments of 2%, as described in the main text.

with 4 or more tendon branches, the joint kinematics showed a notably increased sensitivity for the proposed branching modeling methods. Moreover, the models built using the proposed method were significantly more sensitive than those built using the conventional method.

Regarding the sensitivity to tendon/ligament resting length, within the category of varying numbers of tendon branches (Fig 9A), the proposed method yielded higher Δ values compared to the conventional method, suggesting that the proposed method is more responsive to changes in number of tendon branches. In the category of varying numbers of joints between insertions (Fig 9B), both methods demonstrated similar levels of Δ values, indicating a comparable sensitivity to the number of joints between insertions. In the category of varying muscle fiber length to MTU length ratios (Fig 9C), an increase in the ratio of muscle fiber length to MTU length resulted in a decrease in Δ values for the conventional method. Conversely, the

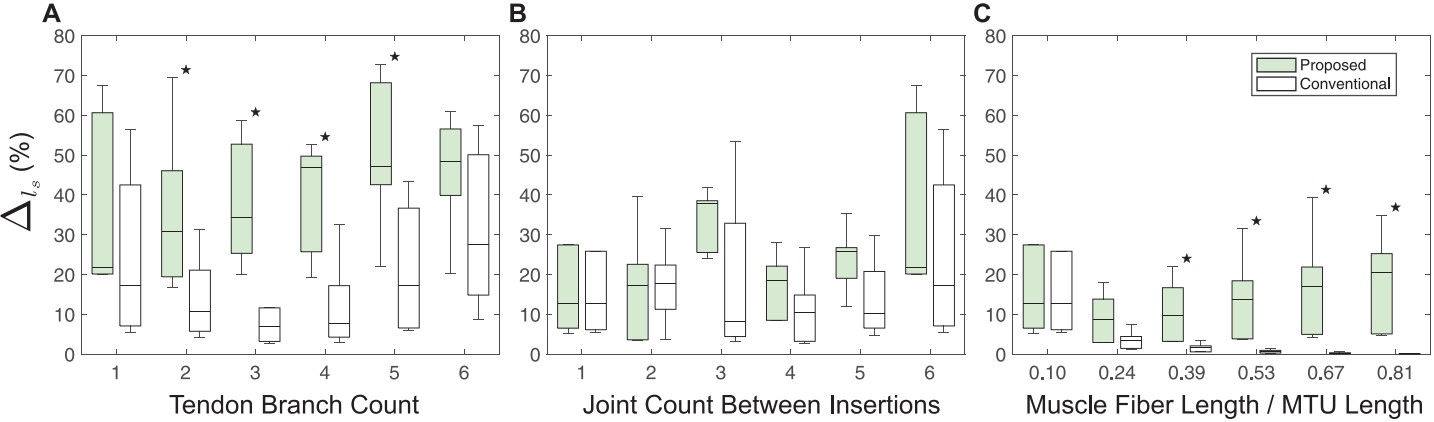

**Fig 9. Ligament resting length sensitivity analysis.** Impact of ligament resting length changes in the proposed versus conventional method on articulated system movement, measured by (2) across variations in tendon branch count (**A**), joint count between insertions (**B**), and muscle fiber length to the MTU length ratio (**C**). Solid black stars, if present, above the box plots indicate that the difference in Δ values between two groups, modeled by the conventional and proposed methods, is significant, determined by a two-tailed T-test (P-value < 0.05). The sample size for each box in the box plot was 6, corresponding to six perturbations of nominal values adjusted from −5% to + 5% in increments of 2%, as described in the main text.

**Table 1. Computational time comparison.**

| Architecture Category | Conventional (sec) | | Proposed (sec) | |
|---|---|---|---|---|
| | Mean | Variance | Mean | Variance |
| Tendon Branch Count | 4.44 | 3.24 | 4.92 | 5.87 |
| Joint Count Between Insertions | 4.06 | 0.90 | 4.29 | 1.46 |
| Muscle Fiber Length / MTU Length | 4.06 | 0.38 | 3.57 | 0.17 |

proposed method exhibited a relativel consistent level of Δ values across different ratios. Additionally, at each ratio, the proposed method showed higher Δ values than the conventional method, highlighting its higher sensitivity to the muscle fiber length to MTU length ratio.

These findings collectively underscore the importance of considering tendon slack length and the complexity of tendon branching when modeling muscle-tendon systems. They also highlight the enhanced responsiveness of the proposed method to key biomechanical parameters, suggesting its potential for facilitating more realistic simulations in biomechanics research.

## Computational time

In the proposed branching modeling method, introducing extra bodies of extremely small mass could potentially extend computational time due to potentially stiffer dynamics. To evaluate computational performance, we compared the computational times of the simulations using the proposed branching modeling method with those using the conventional branching modeling method. The comparison results are documented in Table 1.

In the table, for each muscle-tendon architecture category, we considered all the simulations presented in Figs 5, 8 and 9 to calculate the average computational time required to complete a simulation, each having a duration of $T = 3$ (sec).

Overall, the average computational times were similar between the two branching modeling methods.

Statistically, a significant difference was observed between the methods in the category of the ratio of muscle fiber length to MTU length, determined by a two-tailed T-test (P-value < 0.05). However, there were no significant differences observed between the methods in categories with varying numbers of tendon branches or joint counts between insertions.

In the category of varying numbers of tendon branches, the simulations using the proposed methods took longer than those using the conventional method. This likely suggests that the more extra bodies of extremely small mass were introduced to the systems, the simulation time increased. It is important to note that the effect of these extra bodies on computational time is influenced by their mass. Here, the mass of the extra bodies used was $5 \times 10^{-4}$ kg, which was 0.5% of a single vertebra (i.e., a single cylinder) in the models. Increasing the mass to a larger value, while still keeping it reasonably small relative to the body mass, can mitigate the slowdown effect.

## Availability and future directions

The source code of `ArborSim` is freely and anonymously accessible on GitHub, under the permissive Apache License 2.0, at https://github.com/EMBiRLab/ArborSim.

While the current version of the tool has proven effective by going through the procedure to create the toy models, we recognize opportunities for advancement and refinement. Several functionalities and improvements can be envisioned to extend its applicability and ease of use in the future. For example, in the current implementation of the "**Transformer**", global

landmarks are required to establish joint coordinate systems, representing the axes of these systems. Identifying landmarks that can depict the axes of coordinate systems is indeed a crucial goal. However, acquiring these landmark points often necessitates identifying other biological references as a preliminary step. At present, we assume that this process is completed prior to using the designed tool. Nevertheless, integrating this preliminary step into the tool could further streamline the building process.

Moreover, the current implementation of the "**Brancher**" requires users to input anatomical and physiological parameters. Automatic parameter allocation could be realized by incorporating some predefined assumptions in future iterations of the software. Besides these, other elements such as complex coupled joints in which the spatial transformation might be dependent on other joint coordinates may also be included in future iterations of the tool to represent more complex joint movements observed in biological systems.

In the accompanying study, we conducted comparative simulations using the proposed branching modeling method and the conventional method on toy models. The results revealed large differences in the resulting movement between the models created using these two approaches. We believe that these findings will convince researchers the importance of modeling the branched muscle-tendon networks, an aspect largely neglected in existing research, and motivate further research into branching modeling. Looking forward, we aim to collect necessary material data of real muscle-tendon architectures to validate and refine the proposed branching method. We hope that our work will empower researchers to construct detailed, articulated musculoskeletal models of various systems, particularly those with multiple joints and complex muscle-tendon architectures, and study their contributions to animal motion.

It is possible to revise components, add components, or employ the proposed branching modeling method with existing OpenSim musculoskeletal models using ArborSim. Because one can directly access the "local" data used to build these models, one can format these local data into CSV files, which will then be inputted into the tool for model development. By revising the original parameters or appending additional data to the CSV files, one can choose to convert any existing branched muscles and tendons modeled by the conventional branching modeling method to the proposed branching modeling method.

## Supporting information

**S1 Video. Animated motion comparison: Branch number variation.** Demonstrates the differences in motion using the proposed branching modeling method versus the traditional method across models with varying numbers of tendon branches.
(MP4)

**S2 Video. Animated motion comparison: Joint count variation between insertions.** Compares motion in models with varying joint count between tendon insertions using the proposed method versus the conventional method.
(MP4)

**S3 Video. Animated motion comparison: Muscle fiber length / MTU length variation.** Compares motion in models with varying ratios of muscle fiber length to MTU length using the proposed method versus the conventional method.
(MP4)

## Acknowledgments

The authors would like to thank Brian Umberger, Swithin Razu, Pranav Embar, and Yassar Abdelrahman for helpful discussions regarding this topic.

## Author Contributions

**Conceptualization:** Xun Fu, Jack Withers, Talia Y. Moore.

**Funding acquisition:** Talia Y. Moore.

**Methodology:** Xun Fu.

**Project administration:** Talia Y. Moore.

**Resources:** Juri A. Miyamae, Talia Y. Moore.

**Software:** Xun Fu, Jack Withers.

**Validation:** Xun Fu.

**Visualization:** Xun Fu.

**Writing – original draft:** Xun Fu.

**Writing – review & editing:** Xun Fu, Jack Withers, Juri A. Miyamae, Talia Y. Moore.

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
