## [Decision Letter · Decision Letter 0]

4 Mar 2024

Dear Prof. Moore,

Thank you very much for submitting your manuscript "ArborSim: Articulated, branching, OpenSim routing for constructing models of multi-jointed appendages with complex muscle-tendon architecture" for consideration at PLOS Computational Biology.

As with all papers reviewed by the journal, your manuscript was reviewed by members of the editorial board and by several independent reviewers. In light of the reviews (below this email), we would like to invite the resubmission of a significantly-revised version that takes into account the reviewers' comments.

Following the reviewer's comments, it seems like the current work is promising but requires significant additional work to be accepted. I encourage you to revise and resubmit.

We cannot make any decision about publication until we have seen the revised manuscript and your response to the reviewers' comments. Your revised manuscript is also likely to be sent to reviewers for further evaluation.

Sincerely,

Kaushik Jayaram

Guest Editor

PLOS Computational Biology

Mark Alber

Section Editor

PLOS Computational Biology

Following the reviewer's comments, it seems like the current work is promising but requires significant additional work to be accepted. I encourage you to revise and resubmit.

Reviewer's Responses to Questions

**Comments to the Authors:**

Reviewer #1: This is a solid contribution. It will accelerate the workflow for researchers using OpenSim to model more complex musculoskeletal structures. There is much recent interest in tails, but the extension to fingers and other appendages makes it broadly useful. The text is very clearly written and mostly easy to follow. A couple minor additions would make it a very useful contribution.

The toy model is appropriate at this stage, but it will be nice to see this applied to in vivo data. Getting measurements with which to validate the more refined approach will be challenging, but a nod to this exciting prospect could be made in the discussion?

Abstract:

we found that the number of tendon branches and the number of joints between branches are most sensitive to branching modeling method

these things are the most sensitive amongst which pool of candidates? what other parameters/variables were studied?

reading the author summary, I see that perhaps this is meant to drive home how important the modeling approach to handling branching is, and that ArborSim is good at this or presents a unified approach. perhaps this should be more explicitly stated?

l3 in vivo italicized

L73 - multiple indepedent compartment: so each tendon has it's own separate muscle, instead of all tendons converging on the same muscle?

The example in L82-L86 is convincing.

L124 Can the large differences be compared with any experimental data to determine which method is more accurate? Presumably the more realistic branched method?

L340-L352. It's not clear what the function of the extra spherical body is at the branching point, and why it's necessary. Do the branched tendons stem off of it, like a manifold? Is it rigid, or made of ligament/tendon material? Or is the function of it to create a "wrapping surface" that splits the branches of the tendon apart?

What happens if this spherical body branching points slides onto a wrapping surface? Is that ok? Will they interact physically?

Fig 6 is great, and shows the power of the architecture.

L551-553 great argument for this work.

Figure 7 caption - this needs to say what statistical test was run and what the sample size is, and what the sample population is. T-test, maybe? Are the replicates simulations with different levels of maximal isometric force over the range specified in the main text? Would be nice if the caption made the figure more "stand alone."

Reviewer #2: This paper describes a novel software tool to integrate models with many joints containing complex muscle-tendon architectures into the existing OpenSim biomechanical modeling software. I believe that the work is novel and will be a useful tool for the scientific community that enables different types of biomechanical simulations to be performed. The manuscript is also well-written. However, there are a couple of points that should be addressed prior to publication.

Major concerns:

• The authors detail the difference between their novel muscle-tendon modeling approach compared with traditional OpenSim models with a couple of toy examples, demonstrating differences in resulting motion from identical muscle excitation, sensitivity to model error, and computation time. For the computational timing analysis, the authors present the average computational time across multiple models, but no standard deviations are given. This information should be added, with significance testing between conditions as in Figs 7-8.

• When describing the sensitivity analysis, the authors mention that some of the simulations would fail when applying higher perturbations to ligament resting length. Was this true for the traditional modeling approach or only with the new approach?

Minor concerns:

• Figure 6 should have a legend describing the difference in colors

• Line 421 – what does N.B. mean?

• There are some minor inconsistencies in wording/punctuation in the manuscript that should be checked (ex. “the OpenSim” -> OpenSim in line 379, “Opensim -> OpenSim” in line 483)

Reviewer #3: Pros:

- Excellent idea. Developing detailed musculoskeletal model of animals especially ones other than human is very important and the spirit of this work is inline with doing so.

- The work presents a complete pipeline of going from modeling the mechanics to simulating them by adding to the existing infrastructure of OpenSim

- Detailed and clear methodology section describing all the steps taken in the paper to (i) reproduce and (ii) reuse and (iii) reimplement if needed in other frameworks

- Very good idea of using ligaments as tendons to model splitting in muscles

Major concerns:

- No rigorous discussion on the observations from the experiments. In the results from the comparative simulations, tendon branch count and joint count show a non-linear relationship. This has been attributed to a complex, non-linear relationship in the branched muscle models. There needs to be a more in-depth analysis and reflection on where do these differences come from. Are they a result of initialization?

- No plots or results from the muscle dynamics. Researchers would be interested to know why the kinematics are different between the presented approach and the conventional approach.

- How to validate the current methodology? Any comparisons to the ground truth? It is not enough to provide that there is a difference in kinematics when the models are switched. A stronger evidence needs to provided to prove that the presented mode is indeed better at capturing the dynamics of complex musculature. May be use existing OpenSim models with split tendons and compare with the proposed work?

- It is good that proposed method has the same/similar computational overhead as the conventional method. I was hoping for a code profile to clearly show what parts of the code spend the most time and thereby reflect on how future iterations of the methodology can improve it.

Minor concerns:

- Missing how the overall muscle length is computed

- Model performance under different initial conditions

- Longer simulations to highlight the stability of the proposed methodology

- Smaller animals such as mice have extremely low masses in the tail and are often difficult to simulate them. Using the proposed approach adds additional complexity with near zeros masses to split the tendons. Is it a limitation of OpenSim or of the formulation?

**Have the authors made all data and (if applicable) computational code underlying the findings in their manuscript fully available?**

Reviewer #1: Yes

Reviewer #2: None

Reviewer #3: Yes

PLOS authors have the option to publish the peer review history of their article (what does this mean?). If published, this will include your full peer review and any attached files.

Reviewer #1: No

Reviewer #2: No

Reviewer #3: No
---

## [Decision Letter · Decision Letter 1]

10 Jun 2024

Dear Prof. Moore,

We are pleased to inform you that your manuscript 'ArborSim: Articulated, branching, OpenSim routing for constructing models of multi-jointed appendages with complex muscle-tendon architecture' has been provisionally accepted for publication in PLOS Computational Biology.

Best regards,

Kaushik Jayaram

Guest Editor

PLOS Computational Biology

Marc Birtwistle

Section Editor

PLOS Computational Biology

Please address any remaining comments from the reviews and submit. In particular, I request the authors to specifically address the concern ("Please consider describing or providing a pipeline on how to use existing models in OpenSim with the current tool"

) directly as it could be significant increase the appeal of this manuscript to the readers and community users.

Reviewer's Responses to Questions

**Comments to the Authors:**

Reviewer #1: The authors have done extensive work responding to all three reviews, and the manuscript is of high quality.

Reviewer #2: I thank the authors for their detailed responses to the reviewers. All of my concerns have been addressed.

Reviewer #3: I thank the authors for responding to all the concerns raised in the earlier review. The authors have successfully addressed most of my important concerns that has (hopefully) improved and further reinforced the quality of the paper. Especially, Fig R1

and Fig 7 improved the overall discussion on the influence the proposed modeling changes can influence the simulations.

Please consider describing or providing a pipeline on how to use existing models in OpenSim with the current tool. This will facilitate both testing, validation and development of the tool.

Minor:

- Typo: ForceCoordinateLimitForce -> CoordinateLimitForce

- Line 416 has Fig reference missing

**Have the authors made all data and (if applicable) computational code underlying the findings in their manuscript fully available?**

Reviewer #1: Yes

Reviewer #2: Yes

Reviewer #3: Yes

PLOS authors have the option to publish the peer review history of their article (what does this mean?). If published, this will include your full peer review and any attached files.

Reviewer #1: **Yes: **Andrew J Spence

Reviewer #2: No

Reviewer #3: No

---

## [Editor Report · Acceptance letter]

29 Jun 2024

PCOMPBIOL-D-24-00087R1 

ArborSim: Articulated, branching, OpenSim routing for constructing models of multi-jointed appendages with complex muscle-tendon architecture

Dear Dr Moore,

I am pleased to inform you that your manuscript has been formally accepted for publication in PLOS Computational Biology. Your manuscript is now with our production department and you will be notified of the publication date in due course.

With kind regards,

Katalin Szabo
